# Preparation of Naphthalene-Based Flame Retardant for High Fire Safety and Smoke Suppression of Epoxy Resin

**DOI:** 10.3390/molecules28114287

**Published:** 2023-05-24

**Authors:** Ziqin Huang, Fangli Li, Mingyan Huang, Wenqiao Meng, Wenhui Rao, Yuan Lei, Chuanbai Yu

**Affiliations:** 1College of Materials Science and Engineering, Guilin University of Technology (GUT), Guilin 541004, China; 18776119751@163.com (Z.H.); 18579448271@163.com (F.L.); 18776030125@163.com (M.H.); mengwq@163.com (W.M.); raowh1@163.com (W.R.); 2China Antimony Corporation, Nanning 530001, China

**Keywords:** epoxy resin, thermal performance, flame retardancy, smoke suppression

## Abstract

One of the current challenges in the development of flame retardants is the preparation of an environmentally friendly multi-element synergistic flame retardant to improve the flame retardancy, mechanical performance, and thermal performance of composites. This study synthesized an organic flame retardant (APH) using (3-aminopropyl) triethoxysilane (KH-550), 1,4-phthalaadehyde, 1,5-diaminonaphthalene, and 9,10-dihydro-9-oxa-10-phosphaphenanthrene-10-oxide (DOPO) as raw materials, through the Kabachnik-Fields reaction. Adding APH to epoxy resin (EP) composites could greatly improve their flame retardancy. For instance, UL-94 with 4 wt% APH/EP reached the V-0 rating and had an LOI as high as 31.2%. Additionally, the peak heat release rate (PHRR), average heat release rate (AvHRR), total heat release (THR), and total smoke produced (TSP) of 4% APH/EP were 34.1%, 31.8%, 15.2%, and 38.4% lower than EP, respectively. The addition of APH improved the mechanical performance and thermal performance of the composites. After adding 1% APH, the impact strength increased by 15.0%, which was attributed to the good compatibility between APH and EP. The TG and DSC analyses revealed that the APH/EP composites that incorporated rigid naphthalene ring groups had higher glass transition temperatures (Tg) and a higher amount of char residue (C_700_). The pyrolysis products of APH/EP were systematically investigated, and the results revealed that flame retardancy of APH was realized by the condensed-phase mechanism. APH has good compatibility with EP, excellent thermal performance, enhanced mechanical performance and rational flame retardancy, and the combustion products of the as-prepared composites complied with the green and environmental protection standards which are also broadly applied in industry.

## 1. Introduction

Epoxy resin is an environmentally friendly thermosetting resin [1,2], which is widely used in electronic equipment, aerospace, national defense and the military industry and other industrial applications, due to its advantages of easy processing and good chemical stability [3,4]. Nevertheless, epoxy resin is an ignitable material that generates a lot of heat and smoke when burning, and hence is a considerable fire hazard, thereby limiting its application prospects [5,6,7].

In order to make it safer, the flame retardancy of the epoxy resin must be elevated using appropriate additives [8,9,10,11]. Based on the elements contained, flame retardants can be classified into halogen-free flame retardants and halogen-containing flame retardants [12,13]. Nevertheless, halogen-containing flame retardants produce a large quantity of toxic fumes when burned, which will cause harm to the environment and cause health issues. Hence, the focus of research has shifted to halogen-free flame retardants [14,15]. To improve flame retardancy, it is necessary to add a large amount of halogen-free flame retardants to the epoxy resin (e.g., aluminum hydroxide, aluminum phosphate, ammonium polyphosphate) [16,17]. Nevertheless, halogen-free flame retardants often have disadvantages, such as large quantity requirements leading to a poor dispersion effect which affects the thermal or mechanical performance.

9,10-dihydro-9-oxa-10-phosphaphenanthrene-10-oxide (DOPO) is an environmentally friendly organic P flame retardant that has been widely applied in the production of flame-retarded epoxy resin [18,19,20]. However, the excellent flame retardancy of DOPO can only be achieved at high concentrations, while high DOPO content can lead to the drastic degradation of mechanical and thermal performance of flame-retarded epoxy resins [21]. To ensure a balanced performance, it is necessary to increase the reactive functional groups. Hence, the design and synthesis based on DOPO derivatives have attracted great attention. It is very challenging to combine P, N, and Si based on flame retardant synergies to create an environmentally friendly flame retardant that has a synergy with other properties. Li et al. [22] combined poly(vinyl)silazane (PVSZ) and DOPO through addition, to successfully prepare the phosphorus nitrogen silicon synergistic flame retardant (PPVSZ). When the PPVSZ content was 2.5 wt%, the UL-94 test reached the V-0 rating, the LOI reached 30%, and the impact strength increased by 45.38%, which significantly improved the flame retardancy and mechanical performance of epoxy resin. Unfortunately, this cannot preserve the thermal performance of composites. To improve the thermal performance, rigid groups, such as benzene ring, biphenyl ring, and naphthalene ring, can be introduced into the system to increase the rigidity of the molecular segments and effectively improve the thermal resistance of the epoxy resin curing system [23,24,25]. Gao et al. [26] synthesized the bio-based epoxy monomer diglycidyl ether luteolin (DGEL) from luteolin. The test results showed that the Tg of DGEL/DDS reached 314.4 °C, and the vertical burning test (UL-94) reached the V-0 level, which has both good flame retardancy and good thermal performance. This is because the aromatic rings and multifunctional groups in luteolin improve the thermal stability. By designing novel epoxy resin molecular chains, compounds with rigid groups can be used as monomers to synthesize novel epoxy resin prepolymers and improve the thermal stability of polymers. Although the excellent effect can be obtained in this way, the operation is complicated and it is not amenable to large-scale production. Therefore, realizing excellent mechanical performance, thermal performance, and flame retardancy of phosphazidosilane compounds in EP composites based on the multi-component synergistic effect is a priority [27,28,29].

In this study, a novel P-containing nitrogen silane compound was synthesized; a novel flame retardant, APH, was synthesized by using DOPO, KH550, 1,5-diaminonaphthalene, and terephthalaldehyde via the Kabachnik-Fields reaction. In addition, the quantity of APH added and its effect on the mechanical performance, thermal performance, and flame retardancy of EP composites were analyzed. Analysis of the structure and morphology of products in the thermal decomposition of APH/EP composites, components, and carbon residue revealed the mechanism of flame retardancy of APH in the combustion process.

## 2. Results and Discussion

### 2.1. Material Characterization of APH

The synthetic route of APH is shown in Figure 1. The flame retardant APH was synthesized via the Kabachnik-Fields reaction of KH-550, 1,4-phthalaadehyde, 1,5-diaminonaphthalene, and DOPO. The abundant active amines in APH could participate in the curing reaction of EP, thereby improving the interface compatibility between the flame retardant and EP matrix. Additionally, the introduced naphthalene ring structure could improve the mechanical and thermal performances of the composites. The P–N–Si elements in APH could achieve a synergistic flame-retardant effect, thus endowing the APH/EP composite with high flame retardancy.

Figure 1a showed the FT-IR spectra of APH. As observed, the absorption peak at 2389 cm^−1^ on DOPO corresponds to P-H, the absorption peak of 1695 cm^−1^ on terephthalaldehyde corresponds to C=O, and the characteristic peak at 3376 cm^−1^ on 1,5-diaminonaphthalene is attributed to the N-H stretching vibration of the primary amino group [30]. It was observed that the peak at 2389 cm^−1^ on DOPO and the peak at 1695 cm^−1^ on terephthalaldehyde disappeared at APH, and the double band at 3376 cm^−1^ in 1,5-diaminonaphthalene became a single broad resolution in the APH band [31]. The center was at 3400 cm^−1^ and the absorption peaks of the P–C and P–O–C bonds were at 760 cm^−1^ and 1203 cm^−1^ of APH, respectively, suggesting that aminomethylation occurred during the reaction [32,33]. Figure 1b showed the SEM images of APH. As observed, the APH granule grew with a mushroom structure. Figure 1c,d showed the ^1^H NMR spectrum and ^31^P NMR spectrum of APH. The attribution of each peak in the relevant ^1^H NMR is as follows: δ = 0.33 ppm (q, 2H, –Si–CH_2_–CH_2_–), (labeled “c”), 1.1 ppm (q, 2H, CH_3_–CH_2_–O–Si–), (labeled “a”), 1.22 ppm (q, 2H, –CH_3_–CH_2_–CH_2_–), (labeled “d”), 2.49 ppm (q, 2H, –NH_2_–), (labeled “e”), 3.4 ppm (q, 2H, –CH_2_–CH_2_–NH–), (labeled “f”), 4.35 ppm (q, 2H, CH_3_–CH_2_–O–Si–), (labeled “b”), 7–8 ppm (m, Ar–H), (labeled “h”), 8.26 ppm (d, 1H, CH), (labeled “g”). ^31^P NMR (DMSO-*d*_6_, 400 MHz): δ = 33.5 ppm.

From the above data, the peaks in the range of 8.4–7.0 ppm on the APH in the ^1^H NMR spectrum correspond to the hydrogen signals on the benzene ring. A single peak appeared at 33.5 ppm in the ^31^P NMR spectrum, while the ^31^P NMR spectrum peak of DOPO was located at 13–17 ppm [34]. The combination of FT-TR and NMR results indicated the successful synthesis of APH.

### 2.2. Flame Retardancy

The UL-94 and LOI tests were applied on EP and APH/EP to evaluate the flame retardancy of composites in air. According to Figure 2a,b,d,e and Table 1, the LOI of EP was 23.3%, and the total ignition time (TTI) was 80 s. It did not pass the UL-94 test level, suggesting that EP was flammable in air, and there was a fire safety hazard [35]. As a comparison, after adding the flame retardant, the vertical burning time was shortened and the limiting oxygen index was increased. The LOI of 4% APH/EP was as high as 31.2%, and it reached the V-0 grade, suggesting that the addition of APH markedly improved the flame retardancy of the composites.

Cone calorimeter tests are used to appraise the combustion properties of the polymer. Several key parameters can be obtained through cone calorimetry, for instance, the heat release rate (HRR), smoke release rate (SPR), total heat release (THR), CO production (COP), and total smoke produced (TSP) [36]. The time-varying trends of the above parameters for pure EP and composites containing 2 wt% and 4 wt% APH are displayed in Figure 3a–e, respectively. The relevant data are summarized in Table 2. The PHRR, TSP, and THR of pure EP were as high as 1383.6 kW/m^2^, 61.0 m^2^, and 79.0 MJ/m^2^, respectively, indicating that the combustion ability of EP was very strong [37]. As a comparison, the PHRR of 2% APH/EP and 4% APH/EP was 877.3 and 912.1 kW/m^2^, respectively, which were 36.6% and 34.0% lower than EP, and the TSP and THR were also significantly lower than pure EP. This can be assigned to the fact that APH released a large quantity of P-containing free radicals during the combustion process, captured the free radicals produced, and inhibited the combustion process, suggesting that the addition of APH can effectively reduce the combustion ability of composites. As shown in Figure 3b, the SPR exhibited the same trend as the HRR. Among them, the SPR peak of 4% APH/EP was significantly lower than that of EP. This was probably attributed to the enhanced structural stability of the residual char in the epoxy composites. The dense residual char layer could isolate heat and combustible gases, and reduced the SPR. The percentage of carbon residue (W_700_) increased from 11.8% to 19.4%. This was due to the accumulation of phosphoric acid derivatives generated by APH on the surface of the composites during combustion, promoting the formation of carbon residue and effectively blocking the exchange of heat and volatiles [38]. The increase in the carbon residue content showed that APH has better condensed-phase flame retardancy.

Figure 3e was the COP curve of EP and its composites, and the specific trend was the same as that of HRR and SPR. As depicted in Figure 3e, the COP of EP was 0.051 g/s, and the COP corresponding to the addition of 2 wt% and 4 wt% APH was 0.032 and 0.038 g/s, respectively, which was 37.2% and 25.5% lower than that of pure EP. This can be attributed to the fact that APH forms a dense char layer during combustion, which acts as a physical barrier and prevents the combustible gas from escaping.

Figure 2c,f,g show EP, 2% APH/EP, and 4% APH/EP carbon residue images after the CC test, and Figure 2h–j are their corresponding SEM images. According to Figure 2c, the carbon residue surface of EP was relatively broken, the expansion range was small, and the corresponding SEM images have obvious cracks. According to Figure 2f,g, the surface of the carbon residue after adding APH was relatively complete, firm, and swollen, and combined with SEM diagrams, it can be seen that the addition of APH resulted in the formation of a dense carbon layer on the composites. The above data demonstrated that the addition of APH could effectively enhance the flame retardancy of composites.

### 2.3. Mechanism of Flame Retardancy

Raman spectroscopy and XPS were used to clarify the mechanism of flame retardancy. Figure 4a–c show the Raman spectra of the char residue after the CC tests of EP, 2% APH/EP, and 4% APH/EP samples, respectively. The D band (1343 cm^−1^) and G band (1585 cm^−1^) represent the absorption peaks of disordered carbon and sp_2_ hybridized carbon atoms, respectively [39]. Meanwhile, the carbon residues after combustion were assessed on the basis of the I_D_/I_G_ ratio, which reflects the degree of orderliness in the carbon atoms, to clarify the graphitization level. As shown in Figure 4a, the I_D_/I_G_ ratio of EP was the highest (3.1), while the I_D_/I_G_ ratios of 2% APH/EP and 4% APH/EP were 2.8 and 2.7, respectively, which were lower than those of pure EP. The Raman spectrum indicated that APH can promote the formation of the carbon residue, which is beneficial to enhancing the flame retardancy of the composites [40].

Figure 4d was the XPS full spectrum of carbon residue of 4% APH/EP after the CC test, and the specific composition is shown in Table 3. Figure 4f–i are the O 1s, C 1s, Si 2p, and P 2p spectrums of the carbon residue of 4% APH/EP after the CC test. As indicated by the C 1s spectrum (Figure 4g), the peaks at 286.2 and 288.7 eV belong to the C–O–C bonds and C=O bonds, while the C–C bonds and C=C bonds of the multi-aromatic structures have a peak at 284.8 eV, suggesting that carbon was mainly in the oxidized state at medium and low temperatures. As indicated by the O 1s spectra (Figure 4f), the peaks at 530.5 eV, 531.5 eV, and 532.7 eV were ascribed to PO_3_, P–O–P, and C–O–C/P–O–C, respectively. By comparing the O 1s curves of residual char with the 4% APH/EP composite material (Figure 4f), it was found that the O 1s curves of pure EP (Figure 4e) did not contain phosphorus elements. As indicated by the Si 2p spectrum (Figure 4h), Si–C and Si–O–Si have peaks at 101.9 and 103.1 eV, respectively [41]. As indicated by the P 2p spectrum (Figure 4i), the peaks at 132.8 and 133.4 eV belong to P–O–P/PO_3_ and P–O–C [42]. The above results indicate that APH would generate polyphosphate during the combustion process, facilitate the formation of a dense coke layer, isolate the combustible gas, and exhibit a condensed state mechanism of flame retardancy.

The TG-FTIR test was applied to characterize the combustibles, and the gaseous products released during the combustion of EP were analyzed. Figure 5c,d. showed the FT-IR spectra of the thermal degradation products of EP and APH/EP. As observed, the characteristic peak of the epoxy resin and its composites under different degradation temperatures was basically the same [43]. The absorption peak of water or hydroxyl was located at 3650 cm^−1^; 2900 cm^−1^ corresponds to hydrocarbons, 2350 cm^−1^ corresponds to carbon dioxide, 1710 cm^−1^ represents carbonyl compounds, and aromatic compounds and ethers are at 1520 cm^−1^ and 1150 cm^−1^, respectively [44,45]. As shown in Figure 5d, 4% APH/EP has a characteristic absorption peak of higher intensity in the vicinity of 350 °C, which can be attributed to the degradation of phosphorus oxides in APH. Figure 5e–j show the relationship between the absorption intensity and time of volatiles during the thermal degradation of EP and 4% APH/EP. As observed, after the addition of APH the amount of combustible gas in the thermal degradation of the material dropped drastically. As shown in Figure 5a,b, the difference in gaseous pyrolysis products released by EP and 4% APH/EP were not significant, and Figure 5e displays the fact that the absorption peak of water or hydroxyl is significantly enhanced, suggesting that the flame-retardant process produces more moisture [46]. In combination with the laboratory results of CC, this suggests that it may be caused by the dehydration of APH into carbon. Compared with EP, the intensities of other characteristic peaks were all lower, indicating that the addition of APH reduced the release of organic volatiles such as carbonyl compounds and aromatic compounds in EP composites without significantly changing the gaseous volatiles of composites, so that the APH/EP composite has better flame retardancy and smoke suppression capability [47].

The conception of the principle is shown in Figure 2, which explains the mechanism of the addition of APH to enhance the flame retardancy of EP composites. The synergistic effect of P/N/Si in APH could accelerate the formation of carbon residue. Under the synergistic effect of the P/N/Si compounds, the dense carbon layer formed during APH combustion acted as a physical barrier that could isolate the penetration of oxygen and combustible gas and reduced heat release at the same time, which was mainly manifested in the condensed-phase mechanism of flame retardancy.

### 2.4. Thermal Stability

The thermal stability of epoxy resin-cured products under N_2_ was investigated. Figure 6a shows the TG curves of pure EP and EP with 2 wt% and 4 wt% APH. As shown in Figure 6a, the cured products of the epoxy resin all exhibited a one-step decomposition process. As depicted in Figure 6b, the weight-loss range of the epoxy cured product was from 360 °C to 460 °C, and the amount of carbon residue increased progressively with the increase in the mass of APH. Table 4 summarizes the maximum degradation temperature (T_max_), the initial degradation temperature (T_5%_), maximum mass loss rate (R_Tmax_), and carbon residue (C_700_) of composites at 700 °C. As depicted in Table 4, T_5%_, T_max_, R_Tmax_ all decreased with the increase in APH mass. T_5%_ decreased from 399.4 °C to 389.4 °C. This can be assigned to the fact that during the early decomposition of APH it decomposes into phosphoric acid compounds at high temperature, with poor stability [48]. R_Tmax_ decreased from 1.8%/°C to 1.4%/°C, suggesting that the addition of APH reduced the combustion decomposition rate of the composites and promoted the generation of a coke layer. C_700_ increased from 19.5% to 22.0%, as APH/EP produced a variety of phosphoric acid substances during the decomposition process, which promoted the increase in carbon residue.

The thermal performance of the composites was further explored by using the DSC test. Figure 6c shows the DSC curves of the EP and APH/EP composites. As observed, the glass transition temperature (Tg) of pure EP was 154.4 °C, and when the added APH was 1 wt%, 2 wt%, 3 wt%, and 4 wt%, the glass transition temperature became 155.6 °C, 158.8 °C, 161.9 °C, and 162.7 °C, respectively. The Tg of the composites increased as the amount of added flame retardant increased. This may be because the rigid naphthalene ring groups introduced in APH can increase the rigidity of the molecular chain segment and the compactness of the cross-linked network after the epoxy resin was cured, thus leading to the improvement in the APH/EP composite thermal resistance [49].

### 2.5. Mechanical Performance

Epoxy resin is a commonly used thermosetting resin, and modification for flame retardancy should consider its mechanical performance, especially toughness. Figure 6d shows the impact strength diagram of EP and APH/EP composites. As observed, the impact strength of pure EP was 20 KJ/m^2^; when the added amount was 1 wt%, the impact strength rose to 23 KJ/m^2^, which was 15% higher than that of pure EP. As the amount of added APH increased, the impact strength decreased gradually. When the added amount was 3 wt%, the impact strength dropped to 17.88 KJ/m^2^, which was lower than that of pure EP. This can be assigned to the fact that APH contains many rigid groups, and when the proportion increases the flame-retardant agglomerates in EP, resulting in a decline in the mechanical performance. Figure 6e,f are the SEM images of the impact sections of EP and 1% APH/EP. According to Figure 6e, the impact section of EP is smooth, with flat lines, showing brittle fracture characteristics. According to Figure 6f, the impact section of 1% APH/EP shows wrinkles without agglomeration, and the appearance of these wrinkles effectively disperses the impact force, which is beneficial to the improvement of impact strength.

## 3. Experimental

### 3.1. Materials

Ethanol, 1,4-phthalaadehyde, 1,5-diaminonaphthalene and 3-aminopropyltriethoxysilane (KH550, AR, 98%) were purchased from Sichuan Xilong Science Co., Ltd. (Chengdu, China). Epoxy resin (E51, epoxy value = 0.51) was purchased from Sinopec (Beijing, China). 4,4-Diaminodiphenylmethane (DDM) was purchased from Sichuan Xilong Science Co., Ltd. (Chengdu, China). DOPO was acquired from Guangdong Wengjiang Co., Ltd. (Shaoguan, China). Further purification was not applied for any purchased reagent.

### 3.2. Synthesis of Flame Retardant APH

Firstly, 5.2733 g of 1,5-diaminonaphthalene and 14.5799 g of KH-550 were weighed and added into a 500 mL three-necked flask, and 50 mL of ethanol was added as solvent. The solution was then ultrasonicated for 10 min; 13.4132 g of 1,4-phthalaadehyde was dissolved in 50 mL of ethanol, followed by addition to the as-prepared system. The temperature was set to 60 °C, and the product was condensed and refluxed with magnetic stirring for 5 h. Subsequently, 30.0000 g of DOPO was dissolved in 100 mL of ethanol and the DOPO solution was added to the product. The temperature was raised to 80 °C and maintained for 6 h. After cooling down to room temperature, the solution was filtered and the product was rinsed with ethanol, dried under vacuum at 60 °C for 24 h, ground, and sieved for later use. The product so obtained was denoted as APH.

### 3.3. Preparation of EP and APH/EP Composite

EP (E51) was preheated at 60 °C for half an hour and flame retardant APH and EP were mixed and magnetically stirred at 80 °C for 1 h to obtain a uniformly dispersed mixture. After mixing evenly, the melted curing agent DDM was added and subjected to vacuum until no bubbles were released. Then, the mixture was added to the preheated mold and cured at 80 °C/2 h, 100 °C/2 h, and 130 °C/2 h. The mass of APH was 1 wt%, 2 wt%, 3 wt%, 4 wt%, and 5 wt%, respectively. The ratio of different EP composites is shown in Table 5.

### 3.4. Test Characterization

Characterization using Fourier-transform infrared (FT-IR) spectroscopy (NEXUS 6700 FT-IR spectrometer, Thermo Nicolet, Waltham, MA, USA, wavenumber = 500–4000 cm^−1^), X-ray photoelectron spectroscopy (ESCALAB 250Xi, Thermo Fisher Scientific, MA, USA, 1361 eV Al Ka), scanning electron microscopy (SEM) (S-4800H, Japanese High tech Company, Tokyo, Japan), NMR spectroscopy (^1^H and ^31^P, Bruker AV 400, German company, Karlsruhe, Germany, with deuterated DMSO-*d*_6_ as the solvent), and TGA (TA Q500, TA Company, Newcastle, DE, USA, 35–700 °C under N_2_, 10 °C min^−1^) were applied to the samples. Additionally, the limiting oxygen indexes (LOI) (JF-3, China Jiangning Analytical Instrument Co., Ltd., Nanjing, China) of the samples were obtained, wherein the sample had a size of 130.0 × 6.5 × 3.2 mm^3^.

The UL-94 vertical burning test (YK-3050, Dongguan Youke Equipment Co., Ltd., Dongguan, China) was executed on samples with sizes of 130.0 × 13.0 × 3.2 mm^3^. A cone calorimeter test (MC-3001, FTT Company, East Grinstead, UK, radiation intensity = 35 kW/m^2^) was executed to clarify the combustion behaviors, wherein the sample had a size of 100 × 100 × 3 mm^3^. Raman spectroscopy (DXR, American Thermal Power Company, Milford, MA, USA, 10 s, excitation wavelength = 532.17 nm) was applied to the samples. The unnotched impact test (DR-802B, SANS Company, Shenzhen, China) was executed on samples with a size of 80 × 10 × 4 mm^3^. TG-FTIR (NEZTSCH STA 449 F, Shanghai Fairborn Precision Instrument Co., Ltd., Shanghai, China) tests were also conducted.

## 4. Conclusions

To popularize epoxy resin composites, it is very important to consider both mechanical performance and thermal properties during flame retardancy modification. In this work, a flame retardant APH was designed and synthesized. When the addition of APH was 1 wt%, the impact strength increased by 15.0%, in contrast to that of pure EP. Meanwhile, the rigid groups in APH were beneficial to the maintenance of thermal stability. Analysis of the flame-retarding performance indicated that 4% APH/EP could pass the V-0 grade of vertical combustion, and its LOI could reach 31.2%. Compared with pure EP, the PHRR, THR, and TSP of 4% APH/EP decreased by 34.1%, 15.2%, and 38.4%, respectively. In addition, the presence of APH reduced toxic gas (CO) emissions by 25.5%, and increased carbon residues. Compared with pure EP, the APH/EP composite had excellent mechanical performance, good thermal stability and flame retardancy, and rational smoke suppression. Furthermore, the P/N/Si compounds in APH showed a good synergistic effect, and the dense carbon layer formed during the combustion process served as a physical barrier that could effectively reduce the heat release and combustible gas emissions, mainly exhibiting a condensed-phase flame-retardant mechanism. This study provided a broad prospect for the fire safety application of epoxy resin composites.

## Data Availability

The data presented in this study are available in this manuscript.

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
