# Peer review of "Preparation of Naphthalene-Based Flame Retardant for High Fire Safety and Smoke Suppression of Epoxy Resin"

_molecules, 2023, doi:10.3390/molecules28114287_

Round 1

Reviewer 1 Report

This manuscript describes that a novel multi-element synergistic flame retardant, APH, was synthesized and used to prepare the flame retardant epoxy resin composites. Some interesting results were obtained. It can be published on Molecules, if a few issues are addressed.

1. The names of each reactant should be added in Scheme 1 to facilitate readers' smooth reading.

2. In Line 155, the author explained that the decrease in COP was due to the formation of a charcoal layer, which is incorrect and should be revised.

3. Fig. 3 b shows the SPR diagram, but it is not mentioned in the article. Please add a description of the SPR diagram.

4. The captions of the Figure 4, it is not CC test Raman curves, but rather the Raman test curves of carbon residual after CC testing. The author should make it clear.

5. The authors should highlight Significance, Experiments and Findings for the Abstract.

6. Throughout the manuscript, there are still some errors of grammar and formatting, so authors should carefully modify.

Author Response

Responses to the reviewers’ comments:

Reviewer 1

Comments to the Author

This manuscript describes that a novel multi-element synergistic flame retardant, APH, was synthesized and used to prepare the flame retardant epoxy resin composites. Some interesting results were obtained. It can be published on Molecules, if a few issues are addressed.

  1. The names of each reactant should be added in Scheme 1 to facilitate readers' smooth reading.

Authors' response:

We appreciate your valuable suggestions and remarks. Scheme 1 has been modified as suggested.

  1. In Line 155, the author explained that the decrease in COP was due to the formation of a charcoal layer, which is incorrect and should be revised.

Authors' response:

Thanks for your valuable comments. We apologize for these mistakes. We have modified the charcoal layer to char layer (Line 209). The manuscript has been thoroughly revised as suggested.

  1. Fig. 3 b shows the SPR diagram, but it is not mentioned in the article. Please add a description of the SPR diagram.

Authors' response:

We appreciate the valuable comment. As shown in Figure. 3 (b), the smoke release rate (SPR) exhibited the same trend as heat release rate (HRR). Among them, the SPR peak of 4% APH/EP was significantly lower than that of EP. This was probably attributed to the enhanced structural stability of the residual char in the epoxy composites. The dense residual char layer could isolate heat and combustible gases, and reduced the SPR. The relevant description has been added into the revised manuscript (Lines 195 to 199).

  1. The captions of the Figure 4, it is not CC test Raman curves, but rather the Raman test curves of carbon residual after CC testing. The author should make it clear.

Authors' response:

Thanks for your valuable comments. We apologize for these mistakes. We have corrected it to “Raman test curves of char residual for EP (a), 2% APH/EP (b) and 4% APH/EP (c) after CC test”. The manuscript has been thoroughly revised as suggested.

  1. The authors should highlight Significance, Experiments and Findings for the Abstract.

Authors' response:

We appreciate the valuable comment. The abstract has been revised to highlight Significance, Experiments, and Findings. The revised abstract is as follows:

        One of the current challenges in the development of flame retardants is the preparation of environmentally friendly multi-element synergistic flame retardant to improve the flame retardancy, mechanical performance, and thermal performance of composites. This study synthesized an organic flame retardant (APH) using (3-aminopropyl) triethoxysilane (KH-550), 1,4-phthalaadehyde, 1,5-diaminonaphthalene, and 9,10-dihydro-9-oxa-10-phosphaphenanthrene-10-oxide (DOPO) as raw materials through the Kabachnik-Fields reaction. Adding APH to epoxy resin (EP) composites could greatly improve their flame retardancy. For instance, UL-94 with 4 wt% APH/EP has reached the V-0 rating and has an LOI as high as 31.2%. Additionally, the peak heat release rate (PHRR), average heat release rate (AvHRR), total heat release (THR), and total smoke produced (TSP) of 4% APH/EP were 34.1%, 31.8%, 15.2%, and 38.4% lower than EP, respectively. The addition of APH improved the mechanical performance and thermal performance of composites. After adding 1% APH, the impact strength increased by 15.0%, which was attributed to the good compatibility between APH and EP. The TG and DSC analyses revealed that the APH/EP composites that incorporating rigid naphthalene ring groups had higher glass transition temperatures (Tg) and char residue (C700). The pyrolysis products of APH/EP were systematically investigated and the results revealed that flame retardancy of APH was realized by the condensed phase mechanism. APH has good compatibility with EP, excellent thermal performance, enhanced mechanical performance, rational flame retardancy and the combustion products of the as-prepared composites complied with green and environmental protection standards, which is also broadly applied in industry.

  1. Throughout the manuscript, there are still some errors of grammar and formatting, so authors should carefully modify.

Authors' response:

We appreciate the valuable comment. The manuscript has been revised as suggested.

Reviewer 2 Report

The manuscript under the title: “Preparation of naphthalene-based flame retardant towards high fire safety and smoke suppression of epoxy resin” is in line with «Molecules» journal. This topic is relevant and will be of interest to the readers of the journal. It based on original research. This research has scientific novelty and practical significance. The article has a typical organization for research articles.
Before the publication it requires significant improvements, especially:

  1. The "Introduction" section: it has been proven that the effect of various modifying additives and fillers on the flammability reduction and physical and chemical properties of epoxy polymer composites is determined by many factors: ……. I think the related references should be cited corresponding to each aspect, e.g. (but not limited to these), which will undoubtedly improve the "Introduction" section:
  • Polymer-Plastics Technology and Materials, 59, 874–883, https://doi.org/10.1080/25740881.2019.1698615
  • Polymers 202113(19), 3332; https://doi.org/10.3390/polym13193332
  • J. Compos. Sci. 20237(5), 178; https://doi.org/10.3390/jcs7050178

·        Polymer Composites20204120252035https://doi.org/10.1002/pc.25517

  1. Figure 4f needs to add data for the original EP.
  2. Why, in the study of impact strength and DSC, the amount of APH introduced was 1, 2, 3, and 4%, while in the study of fire resistance and other properties, the content of APH was 2 and 4%? Why wasn't the effect of 1% APH on other properties investigated, since such a content had the maximum effect on toughness. Why wasn't the amount of APH increased by more than 4%, because the decrease in impact strength was insignificant, and the fire-retardant effect could be even higher.
  3. Section 2 and section 3 need to be swapped.
  4. Section 3.2. How was the grinding of the finished product? What are the dimensions and size distribution of the resulting APH?
  5. Section 3.4. Equipment - you must specify the type, brand, city and country of manufacture for each type of equipment.

Author Response

Responses to the reviewers’ comments:

Reviewer 2

Comments to the Author

The manuscript under the title: “Preparation of naphthalene-based flame retardant towards high fire safety and smoke suppression of epoxy resin” is in line with 《Molecules》 journal. This topic is relevant and will be of interest to the readers of the journal. It based on original research. This research has scientific novelty and practical significance. The article has a typical organization for research articles.

Before the publication it requires significant improvements, especially:

  1. The "Introduction" section: it has been proven that the effect of various modifying additives and fillers on the flammability reduction and physical and chemical properties of epoxy polymer composites is determined by many factors: ……. I think the related references should be cited corresponding to each aspect, e.g. (but not limited to these), which will undoubtedly improve the "Introduction" section:

Polymer-Plastics Technology and Materials, 59, 874-883, https://doi.org/10.1080/25740881.2019.1698615

Polymers 2021, 13(19), 3332; https://doi.org/10.3390/polym13193332

  1. Compos. Sci. 2023, 7(5), 178; https://doi.org/10.3390/jcs7050178

Polymer Composites. 2020; 41: 2025-2035. https://doi.org/10.1002/pc.25517

Authors' response:

We appreciate the valuable comment. The references have been cited in the revised manuscript as suggested (No 11, 24, 25, 29 of References). Please see the revised manuscript.

  1. Figure 4 f needs to add data for the original EP.

Authors' response:

Thanks for your valuable comments. We have added the curve of O 1s for pure EP (Figure. 4 (e)). By comparing the O 1s curves of residual carbon with the 4% APH/EP composite material (Figure 4 f), it was found that the O 1s curves of pure EP did not contain phosphorus elements. The relevant description has been added into the revised manuscript (Lines 244 to 246). The manuscript has been thoroughly revised as suggested.

  1. Why, in the study of impact strength and DSC, the amount of APH introduced was 1, 2, 3, and 4%, while in the study of fire resistance and other properties, the content of APH was 2 and 4%? Why wasn't the effect of 1% APH on other properties investigated, since such a content had the maximum effect on toughness. Why wasn't the amount of APH increased by more than 4%, because the decrease in impact strength was insignificant, and the fire-retardant effect could be even higher.

Authors' response:

Thanks for your valuable comments. As shown in Figure 6 (c) and (d), it can be seen that when the addition amount of APH is 1%, 2%, 3%, and 4%, the impact strength test and DSC test results showed significant changes with the change of addition amount. As shown in DSC test, the Tg of the composites increased as the amount of added flame retardant increased. In the impact strength test, the impact strength of APH/EP composite material showed a decreasing trend with the increase of addition amount of flame retardant-. Meanwhile, we conducted UL-94 testing on APH/EP composite materials. In the UL-94 experiment, the flame-retardant properties of composite materials with four ratios of 1, 2, 3, and 4% were tested. The results showed that when the addition amounts were 1% and 2%, the flame-retardant epoxy composites passed the V-1 rating tested by UL-94. When the addition amounts were 3% and 4%, the flame-retardant epoxy composites passed the V-0 rating. The experimental results indicated that the addition of 1% APH and 2% APH or 3% APH and 4% APH had tiny effect on the flame retardancy of the composite material. Hence, on this basis, we selected 2% and 4% points for the next step of research.

The experimental results showed that although the impact strength of the composite material increased significantly when 1% APH was added, the improvement in its thermal and flame-retardant properties was not significant. Therefore, we did not conduct CC and TG-IR tests on 1% APH/EP composite material.

As shown in Figure 2 (d) and (e), it was seen that when the addition amount of APH was 3%, although the composite material passed the V-0 rating of UL-94 testing, however, the time for the twice ignition to extinguish was exactly 10 s. Therefore, the addition amount of APH was further increased. When the addition amount was 4%, the composite material passed the V-0 rating of UL-94 testing, and extinguished within 7 s for twice ignition. UL-94 testing is an important means to determine the combustion performance of EP composite materials. When the addition amount was 4%, the UL-94 experiment has achieved the best results. Meanwhile, considering the effect of 4% addition on the mechanical properties of composite materials, we did not further improve the addition amount of flame retardants.

  1. Section 2 and section 3 need to be swapped.

Authors' response:

Thanks for your valuable comments. The manuscript has been thoroughly revised as suggested.

  1. Section 3.2. How was the grinding of the finished product? What are the dimensions and size distribution of the resulting APH?

Authors' response:

Thanks for your valuable comments. Figure 1 (a) showed a digital photo of flame retardant APH, which could be seen as a delicate solid powder. After synthesizing APH, we ground it and screened the APH solid using a 100 mesh (pore size 0.15 mm) inspection sieve. And scanning electron microscopy testing was conducted on APH, the results showed that the diameter of APH was about 10 μm. Nevertheless, the size distribution of APH was uneven. Meanwhile, APH was a reactive flame retardant that exhibited a molten state when mixed with epoxy resin and the amino groups on the molecular chains will react with the epoxy group. Therefore, the size distribution does not have a significant impact on its performance.

  1. Section 3.4. Equipment-you must specify the type, brand, city and country of manufacture for each type of equipment.

Authors' response:

Thanks for your valuable comments. More information of the equipment has been added into the manuscript: Characterization by using Fourier transform infrared (FT-IR) spectroscopy (NEXUS 6700 FT-IR spectrometer, Thermo Nicolet, USA, wavenumber = 500-4000 cm-1), X-ray photoelectron spectroscopy (ESCALAB 250Xi, Thermo Fisher Scientific, USA, 1361 eV Al Ka), scanning electron microscopy (SEM) (S-4800H, Japanese High tech Company), NMR spectroscopy (1H and 31P, Bruker AV 400, German company, with deuterated DMSO‑d6 as the solvent), TGA (TA Q500, TA Company, USA, 35-700 °C under N2, 10 °C min-1) were applied to the samples. Additionally, the limiting oxygen indexes (LOI) (JF-3, China Jiangning Analytical Instrument Co., Ltd) of the samples were obtained, wherein the sample had a size of 130.0 × 6.5 × 3.2 mm3. The UL-94 vertical burning test (YK-3050, Dongguan Youke Equipment Co., Ltd) was executed on samples with sizes of 130.0 × 13.0 × 3.2 mm3. A cone calorimeter test (MC-3001, FTT Company, UK, radiation intensity = 35 kW/m2) was executed to clarify the combustion behaviors, wherein the sample had a size of 100 × 100 × 3 mm3. Raman spectroscopy (DXR, American Thermal Power Company, 10 s, excitation wavelength = 532.17 nm) was applied to the samples. The unnotched impact test (DR-802B, SANS Company, Shenzhen, China) was executed on samples with a size of 80 × 10 × 4 mm3. TG-FTIR (NEZTSCH STA 449 F, Shanghai Fairborn Precision Instrument Co., Ltd., China) tests were also conducted. (Lines 114 to 130, Section 2.4). The manuscript has been thoroughly revised as suggested.

Round 2

Reviewer 2 Report

The authors considered most of the comments or adequately responded to the remarks contained in the review; therefore, the work may be approved for publication.